# Beneath the surface: Amino acid variation underlying two decades of dengue virus antigenic dynamics in Bangkok, Thailand

Angkana T. Huang[1,2], Henrik Salje[1,3], Ana Coello Escoto[1,4], Nayeem Chowdhury[1], Christian Chávez[1], Bernardo Garcia-Carreras[1], Wiriya Rutvisuttinunt[5], Irina Maljkovic Berry[5], Gregory D. Gromowski[5], Lin Wang[3], Chonticha Klungthong[2], Butsaya Thaisomboonsuk[2], Ananda Nisalak[2†], Luke M. Trimmer-Smith[1], Isabel Rodriguez-Barraquer[6], Damon W. Ellison[5], Anthony R. Jones[2], Stefan Fernandez[2], Stephen J. Thomas[7], Derek J. Smith[8], Richard Jarman[5], Stephen S. Whitehead[9], Derek A. T. Cummings[1]*, Leah C. Katzelnick[1,4]*

1 Department of Biology and Emerging Pathogens Institute, University of Florida, Gainesville, Florida, United States of America, 2 Department of Virology, Armed Forces Research Institute of Medical Sciences, Bangkok, Thailand, 3 Department of Genetics, University of Cambridge, Cambridge, United Kingdom, 4 Laboratory of Infectious Diseases, National Institute of Allergy and Infectious Diseases, National Institutes of Health, Bethesda, Maryland, United States of America, 5 Viral Diseases Branch, Walter Reed Army Institute of Research, Silver Spring, Maryland, United States of America, 6 School of Medicine, University of California, San Francisco, San Francisco, California, United States of America, 7 State University of New York Upstate Medical University, Syracuse, New York, United States of America, 8 Department of Zoology, University of Cambridge, Cambridge, United Kingdom, 9 Laboratory of Viral Diseases, National Institute of Allergy and Infectious Diseases, National Institutes of Health, Bethesda, Maryland, United States of America

† Deceased.
* datc@ufl.edu (DATC); leah.katzelnick@nih.gov (LCK)

**Data Availability Statement:** The authors confirm that all data underlying the findings are fully available without restriction. Data and code for the

## Abstract

Neutralizing antibodies are important correlates of protection against dengue. Yet, determinants of variation in neutralization across strains within the four dengue virus serotypes (DENV1-4) is imperfectly understood. Studies focus on structural DENV proteins, especially the envelope (E), the primary target of anti-DENV antibodies. Although changes in immune recognition (antigenicity) are often attributed to variation in epitope residues, viral processes influencing conformation and epitope accessibility also affect neutralizability, suggesting possible modulating roles of nonstructural proteins. We estimated effects of residue changes in all 10 DENV proteins on antigenic distances between 348 DENV collected from individuals living in Bangkok, Thailand (1994-2014). Antigenic distances were derived from response of each virus to a panel of twenty non-human primate antisera. Across 100 estimations, excluding 10% of virus pairs each time, 77 of 295 positions with residue variability in E consistently conferred antigenic effects; 52 were within ±3 sites of known binding sites of neutralizing human monoclonal antibodies, exceeding expectations from random assignments of effects to sites (p = 0.037). Effects were also identified for 16 sites on the stem/ anchor of E which were only recently shown to become exposed under physiological conditions. For all proteins, except nonstructural protein 2A (NS2A), root-mean-squared-error (RMSE) in predicting distances between pairs held out in each estimation did not outperform sequences of equal length derived from all proteins or E, suggesting that antigenic signals

analyses is held in Zenodo (https://doi.org/10.5281/zenodo.5615512).

**Funding:** This research was supported by the Intramural Research Program of the National Institute of Allergy and Infectious Diseases (SSW, LCK), National Institute of Allergy and Infectious Diseases and National Institutes of Health Grant R01AI114703-01 (https://www.nih.gov, ATH, HS, ACE, NC, CC, BGC, WR, IMB, GDG, LW, CK, BT, AN, LMT, DWE, ARJ, SF, DJS, RJ, DATC, LCK), the Military Infectious Disease Research Program (https://midrp.amedd.army.mil, ATH, WR, IMB, GG, RJ), and a European Research Council Grant 804744 (https://erc.europa.eu, HS). Sequencing for infectious disease surveillance was additionally supported by the Global Emerging Infections Surveillance (GEIS) Branch (https://www.health.mil/Military-Health-Topics/Combat-Support/Armed-Forces-Health-Surveillance-Branch/Global-Emerging-Infections-Surveillance-and-Response, RJ). The funders had no role in study design, data collection and analysis, decision to publish, or preparation of the manuscript.

**Competing interests:** The authors have declared that no competing interests exist. Author Ananda Nisalak was unable to confirm their authorship contributions. On their behalf, the corresponding author has reported their contributions to the best of their knowledge.

present were likely through linkage with E. Adjusted for E, we identified 62/219 sites embedding the excess signals in NS2A. Concatenating these sites to E additionally explained 3.4% to 4.0% of observed variance in antigenic distances compared to E alone (50.5% to 50.8%); RMSE outperformed concatenating E with sites from any protein of the virus ($\Delta$RMSE, 95%IQR: 0.01, 0.05). Our results support examining antigenic determinants beyond the DENV surface.

## Author summary

Dengue viruses, even of the same serotype, are differentially recognized by preexisting antibodies of individuals. With antibody levels being an important indicator of infection risk and pathogenicity, understanding mechanisms underlying these differences are crucial for vaccine design and development. Investigations have primarily targeted surface regions of the envelope protein (E) where virus-antibody interactions were thought to primarily occur. However, the roles of non-surface regions of the E protein as well as non-structural proteins has been limited. We looked at the entire virus to identify associations between specific changes in the protein sequence and differences in how viruses were recognized by antibodies. In addition to recovering known determinants on the surface, we found signals in other areas on the structural building blocks of the virus. We also identified additional signals on specific areas of a protein that does not form structures of the virus but orchestrate virus formation. Our results point towards broadening the frame of investigation to gain a more comprehensive understanding of mechanisms giving rise to antibody recognition of dengue viruses, and may aid the design and evaluation of vaccines and/or assays to characterize dengue immunity.

## Introduction

Dengue virus (DENV) is a vector-borne flavivirus with four recognized serotypes, DENV1–4, which circulate in the human population and cause a spectrum of disease ranging from mild to life-threatening. Anti-DENV immunity is complex and imperfectly understood. The long-standing belief is that infection by a strain of one serotype induces long-term protection against the homologous serotype but only protects against other serotypes for a limited amount of time [1, 2]. However, evidence of antigenic heterogeneity within and among the four DENV serotypes challenges this long-standing belief [3]. Instances of reinfection with strains that exhibit reduced neutralization, albeit rare, have also been reported [4]. As neutralizing antibodies are the best supported correlates of protection [5–8], understanding the determinants of these differences is crucial for vaccine design and implementation.

The DENV genome encodes three structural proteins, the envelope protein (E), precursor membrane protein (prM), and the capsid protein (C), and seven non-structural proteins (NS1, NS2A, NS2B, NS3, NS4A, NS4B, NS5), Fig 1. Anti-DENV antibodies primarily target epitopes on the E protein, prM protein, and the secreted NS1 protein [9, 10]. Previous studies have identified numerous amino acid differences in specific epitopes that modulate neutralization of genotypes within serotypes by both monoclonal antibodies and polyclonal serum antibodies [11–15]. In addition to directly impacting the recognition of epitopes by antibodies, amino acid changes can also affect other viral processes, imposing large effects on the antigenicity of the virus. For instance, a single change on the envelope protein of DENV4 genotype V disrupts

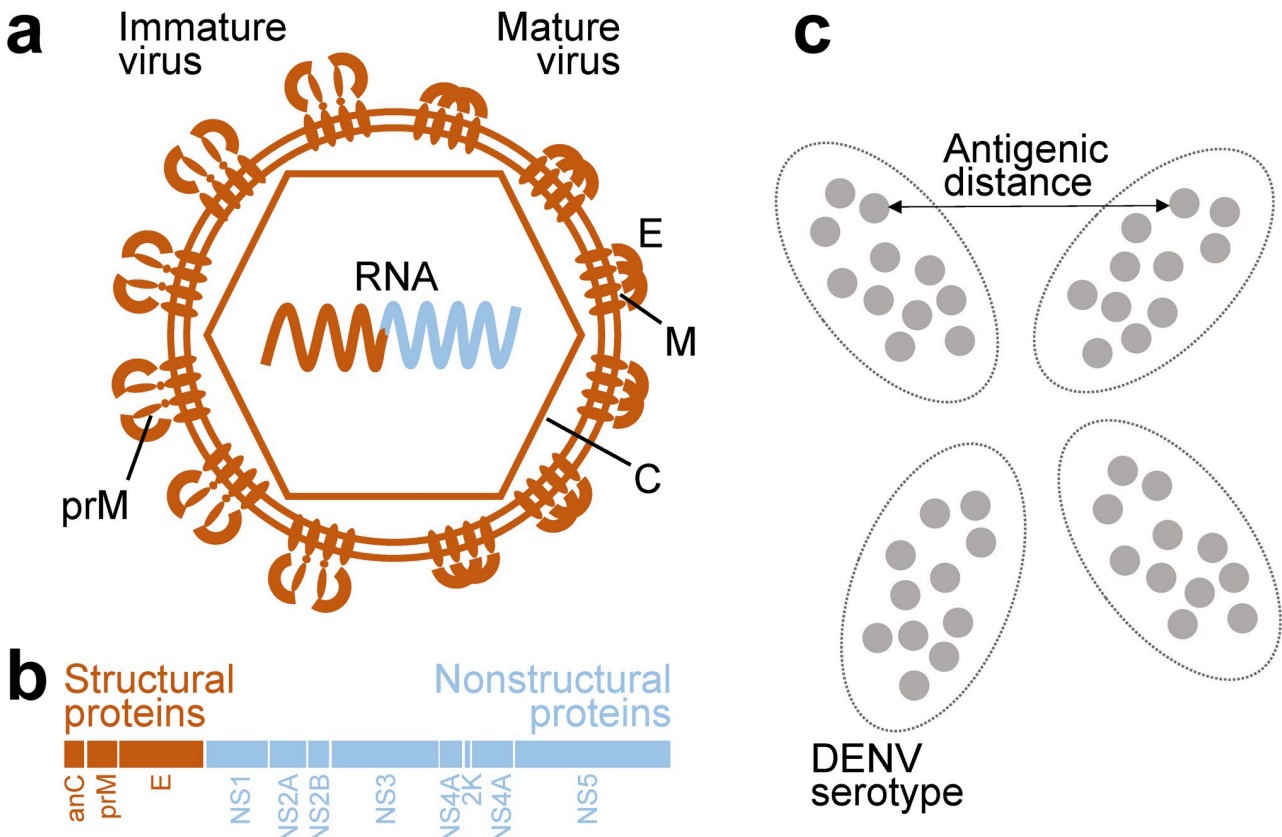

**Fig 1. Dengue proteins and antigenicity.** a) Structure of immature (left-half) and mature (right-half) dengue virion with viral RNA encapsulated. b) Organization of dengue proteins on a polyprotein tranlated from the viral RNA. c) Representation of two-dimensional antigenic map of dengue viruses. Viruses in each of the serotypes form antigenic clusters on the map. Antigenic distances can be measured from the map.

the glycosylation motif of the virus. This change both reduces cell infectivity and hinders the binding and neutralization of some monoclonal antibodies [16]. In addition to increasing epitope heterogeneity, variation in prM cleavage was shown in DENV2 to also affect its structural kinetics. Both mechanisms led to differences in antibody recognition [17]. A mutation which alters the structural kinetics introduced into DENV1 made cryptic epitopes on the E protein more accessible, increasing sensitivity to neutralization [18]. As such, residue changes in non-epitope sites may also modulate DENV antigenic properties more broadly but have yet to be studied in detail.

Numerous previous studies have identified the antigenic effects of substitutions between serotypes [19], between genotypes [15, 20], or in highly lab-adapted viruses [21]. To our knowledge, there are no previous studies that have examined amino acid variation that arises among closely related lineages co-circulating in the same location over time, where antigenic evolution is likely to be most evident. Thus, while previous studies have identified key regions associated with antigenic effects for one serotype or another, little is known about the positions that are variable and result in antigenic shifts both within and between serotypes, especially those that emerge and disappear naturally during circulation. Comprehensive antigenic characterization with a robust serological assay of a large number of closely related, co-circulating strains enables identification of these more subtle antigenically important sites that might otherwise be missed or overlooked. Further, dense sampling of highly similar strains paired with matched full genome sequences enables screening for amino acid changes outside major

epitopes that may have subtle but significant antigenic effects. Such sites are not generally studied in smaller experimental studies of the effects of specific amino acid changes on structural proteins.

To identify associations of antigenic signal with specific substitutions across all DENV proteins, we leveraged a dataset of paired whole genome sequences and neutralizing antibody titers that includes 348 DENV1–4 strains co-circulating in Bangkok, Thailand over two decades (1994 to 2014, S1 Fig) [22]. Viruses were placed onto a 3-dimensional antigenic map using antigenic cartography, a method which fits the Euclidean distance between virus-serum pairs on a map to the difference in titer measurements across a panel of reference sera [3]. Antigenic distances between virus pairs were calculated from these coordinates. We implemented a method that was previously used to estimate effect sizes of substitutions in the E protein on antigenic distances between 47 global DENV1–4 strains [23] on this new dataset. These genetically similar but antigenically diverse viruses allowed us to identify antigenic determinants within serotypes and even within genotypes. In addition to substitutions that were congruent with known epitopes on the E protein, we identified other substitutions that resided in the stem/anchor of E and in nonstructural proteins. After removing substitutions that harbored signals due to co-ancestry with antigenic determinants in E, we found that the remaining substitutions were associated with antigenic effects above that expected by chance. We describe the positions of these substitutions on the protein and discuss their potential role in modulating viral antigenicity. Finally, we probe our virus set for natural experiments to test for observable antigenic effects of individual residues.

## Results

### Substitutions in E are associated with antigenic variation

We applied a similar substitution model to that described in Bell, et al. [23] to estimate the effects of observed amino acid changes on antigenic distances in the Thai virus dataset. The model assumes that distances observed between virus pairs were additive effects of amino acid substitutions separating them. Virus-specific intercepts were added to account for the contributions of virus-specific measurement uncertainties. Biologically, the intercept quantifies the expected distance between two independent characterizations of the same virus. Estimations were done 100 times with a random 10% of the 120,756 virus pairs held out each time. Substitutions which showed effects in at least 95% of the estimations were deemed antigenically relevant.

Using E protein sequences as input, distributions of estimated virus-specific intercepts were similar across the 100 estimations (S2 Fig), with an average of 0.74-fold titer reduction (95% IQR: 0.74, 0.75) across all viruses. This average is similar to the expected antigenic distance resulting from variability of PRNT50 measurements (95%IQR: 0.74, 0.75).

Using the effect size estimates and the average intercept in each fold, we predicted antigenic distances between all virus pairs in the dataset. The predictions of antigenic distances from models fitted with the E protein substitutions showed a tight correlation with the observed distances (correlation coefficient of 0.87, S3 Fig). Benchmarked against predictions assuming distances between centroids of serotypes, the substitutions in E explained 50.5% to 50.9% of the residual variance. To quantify how much within serotype variations were captured, we restricted the calculation to only intraserotype distances and found that 79.1% to 79.3% of the variance was explained.

### Substitution model captures known epitopes in E

The model identified 394 nonzero effect substitutions positioned on 77 of the 295 sites on the E protein with residue diversity observed in the Thai DENV dataset (Fig 2 and S4 Fig, and

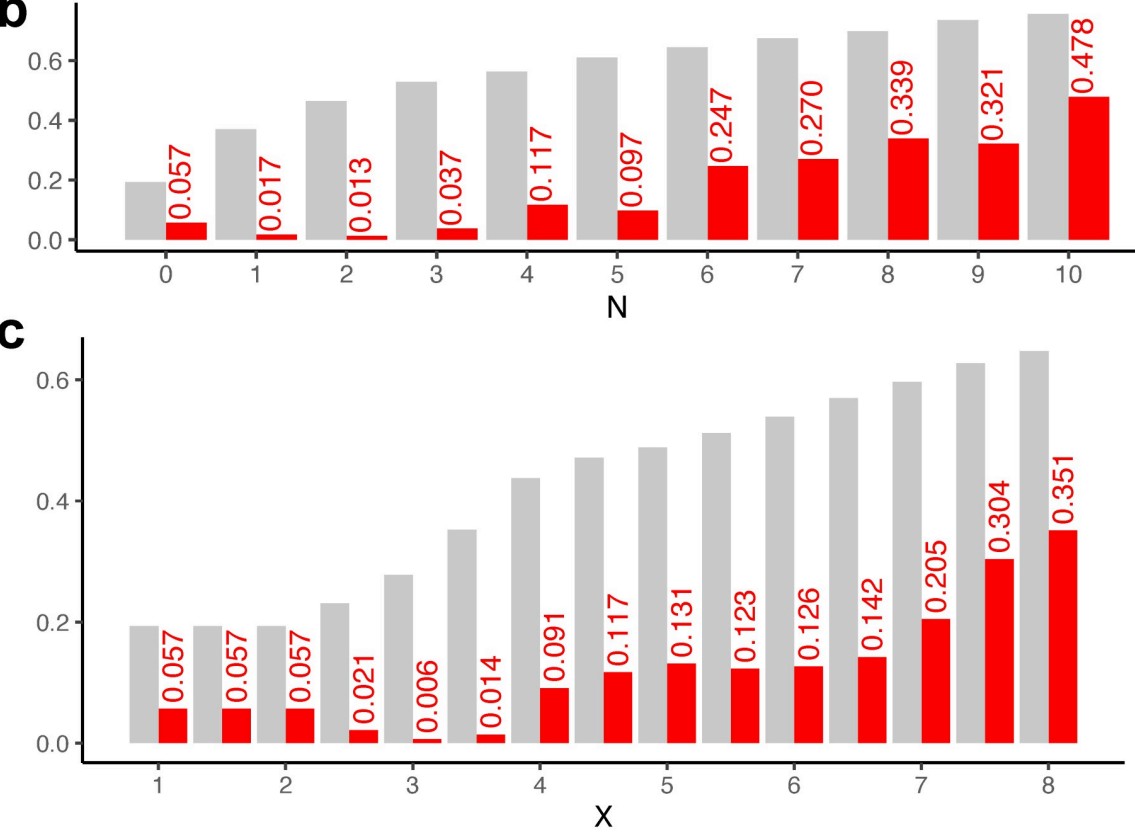

**a**

| | Epitope association | | | |
|---|---|---|---|---|
| | **hmAb + mmAb** | **hmAb** | **Not known** | **Total** |
| **Nonzero effect** | **22 (28.6%)** | **21 (27.3%)** | **55 (71.4%)** | 77 |
| **Zero effect** | **52 (23.9%)** | **36 (16.5%)** | **166 (76.1%)** | 218 |
| **No diversity** | 37 | 30 | 163 | 200 |
| **Total** | 111 | 87 | 384 | 495 |
| **Odds ratio (hmAb + mmAb)   : 1.28 (95%CI: 0.71, 2.29)** | | | | |
| **Odds ratio (hmAb only)   : 1.90 (95%CI: 1.02, 3.51)** | | | | |

Proportion of sites with variability within neighborhood of known epitopes

Probability of overlap being >= the observed by coincidence

**Fig 2. Association between effect sites and known epitopes of neutralizing antibodies.** a) Number and percentage of sites with and without effects by whether or not they are part of known epitopes. Odds ratios were calculated by either considering epitopes of both human-derived monoclonal antibodies (hmAb) and murine-derived monoclonal antibodies (mmAb) and when only restricted to hmAb epitopes. b) Defining neighborhoods of known hmAb epitopes as positions within N sites away (linear distance), the probability of nonzero effect sites being within the neighborhood at random (red) are contrasted against the proportion of variable sites that were

within the neighborhood (gray). c) Analogous analysis but with neighborhoods defined as being within X angstroms away from known hmAb epitopes (3-dimensional spatial distance). N = 0 and X = 0 were when the neighborhood was exactly at the reported epitope positions.

S1 File). The number of substitutions identified and number of sites involved changed minimally when we only considered effects present in 100% of estimations as antigenically relevant (S5 Fig). To evaluate whether the nonzero effect size positions were in epitopes previously identified for anti-DENV antibodies, we compared the positions against those compiled in a comprehensive database of DENV monoclonal antibodies (DENVab) [24]. This database includes information on the identified sites which constituted the epitopes (footprints) for 253 anti-DENV human and mouse monoclonal antibodies (mAbs) that were reported in the literature up to 2016, including potently neutralizing antibodies that have been characterized previously [25–27].

According to DENVab database, 159 positions in the E protein contribute to epitopes of characterized anti-DENV mAbs while 336 positions have not yet been associated with any epitopes. Seventy of the mAbs were recorded to have neutralization activity, footprints involving 111 sites. Of these, 74 sites were variable in our dataset meaning their effects have the potential to be detected by the model. Of the 77 amino acid positions identified as having non-zero effect sizes by our model, 22 (28.6%) were within these footprints, while 55 (71.4%) were not (Fig 2 and S4 Fig). The odds ratio for nonzero effects being in known epitopes versus not known sites was not significantly different from one (1.28, 95%CI: 0.71, 2.29). Restricting the assessment to footprints of human-derived mAb (hmAb) that were variable in our dataset (57 sites) revealed a significant positive association (odds ratio of 1.90, 95%CI: 1.02, 3.51); 21/77 (27.3%) fell within hmAb footprints. Well characterized binding sites of potently neutralizing type-specific hmAb 1F4, 14C10, 2D22, and 5J7, as well as broadly cross-neutralizing mAbs EDE1–2B2 and EDE1–2C8 were captured (Fig 3).

Interestingly, while 36 sites previously identified as DENV-specific hmAb epitopes were marked as zero-effect size by the model, 29 sites (80.5%) were within 3 linear positions away from a nonzero effect residue. In reverse, of the 56 nonzero effect sites that did not match the reported hmAb epitopes, 28 were within 3 sites of known hmAb epitopes, suggesting that they plausibly could contribute to epitopes for some previously identified antibodies. The chance of observing at least this amount of overlap, 21 captured + 28 proximal, if 77 sites were chosen from the 295 sites with variability at random was small (p = 0.037, Fig 2B). We repeated the analysis using distances extracted from a resolved 3-dimensional structure of E [28]. The chance was also small when proximal sites were defined as being within 3.5 angstroms away (p = 0.014, Fig 2C).

Sixteen of the sites identified by the model to have antigenic signals were located in the stem/anchor domain, sites that were unlikely targets of antibodies and were not within or near the sites of previously identified epitopes. However, recent conformational studies have revealed large increases in accessibility to sites on the amphipathic stem helices (E:431–448) and the transmembrane helices (E:465–486) when temperature was heightened from 28°$C$ to 37°$C$ (DENV2) and 40°$C$ (DENV1) [29] which may allow these cryptic epitopes to be exposed under these physiological conditions.

## Antigenic signals exist in proteins beyond E

Aside from sites on the exposed DENV proteins, non-antigenically relevant sites can also harbor antigenic signals if they are linked to antigenic sites. Phylogenies inferred from individual genes were shown to have branching patterns similar to ones inferred from the complete

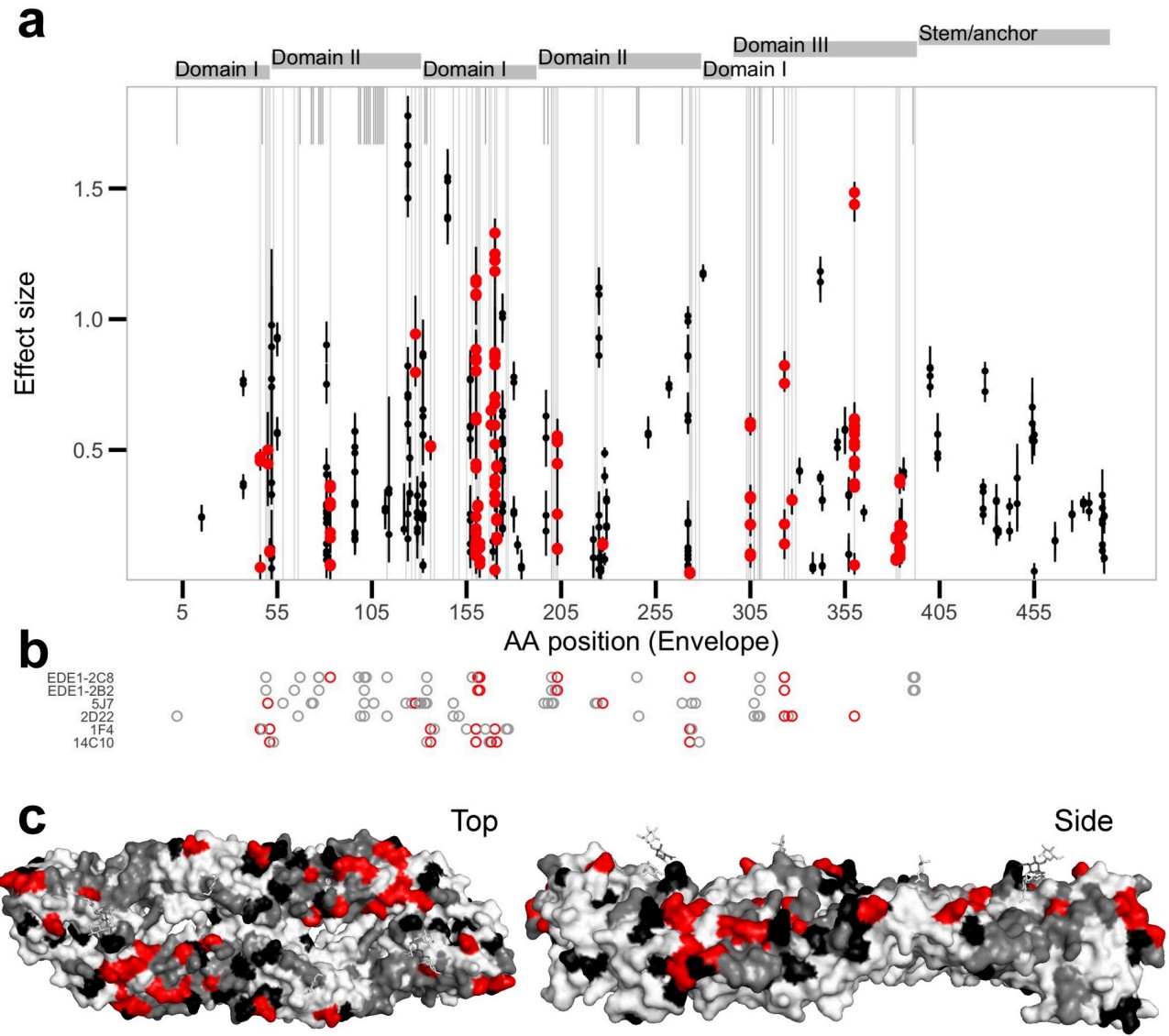

**Fig 3. Effects of substitutions in the envelope protein.** a) Substitutions with non-zero effect sizes with 95% interquartile range across the 100-fold Monte Carlo cross-validations as whiskers, median as points. Points are colored red if they match positions of known epitopes for monoclonal antibodies compiled in the DENVab database [24]. Gray vertical lines indicate positions with known human-derived monoclonal antibody (hmAb) epitopes, long if within site diversity exists in our dataset and short if not. b) Footprints of potently neutralizing hmAbs, colored red if the positions showed non-zero effects. c) Top and side views of the envelope protein structure with known epitopes colored red if estimated as non-zero effect, and gray if estimated as zero effect. Non-zero effect positions not matching reported hmAb epitopes are in black.

genome or the open reading frame (ORF), with nonstructural genes, except NS4A, yielding better resolution (i.e., stronger clade support values) than structural genes [30–32] despite some reports of DENV intraserotype recombination [33]. Hence, it is not unexpected that correlated amino acid changes in other proteins, as a result of correlated nucleotide changes due to shared ancestries, would appear as predictive. However, if the association with antigenic signals were only due to associations with other sites, the prediction performance should be bounded at most by the performance of the signal contributing protein and decline as the linkage dissolves. To identify antigenic determinants in proteins other than E, (1) we fitted effect sizes for each of the DENV proteins separately, (2) screened for proteins with predictive

performance exceeding that of sites in E, (3) sifted for sites that were consistently associated with antigenic signals when adjusted for sites in E, and (4) showed that the signals in these sites were not coincidentally mapped to antigenic variation at random.

To screen for proteins with more antigenic signal than expected from linkage alone, performance of antigenic distance predictions for each of the DENV proteins, measured as root mean squared error (RMSE), were compared against 20 random subsamples of the polyprotein (sampled to the length of each respective protein). This adjusts for protein size: as seen in Fig 4, larger proteins are associated with better prediction performances (lower RMSE). Most proteins showed equivalent performance to the random samples from the polyprotein or worse (95%IQR of the RMSE overlapping or higher than 95%IQR of the comparator) except for NS2A which on average outperformed the comparator by 0.04 (0.02, 0.06) in absolute difference in RMSE. A similar analysis but subsampling from sites on the E protein (rather than the entire polyprotein) was performed to screen for proteins harboring signals beyond expected from covariations with sites in E. Again, only NS2A had a lower RMSE (and non-overlapping 95%IQR) than expected: RMSE difference of 0.03 (0.01, 0.06) from sites sampled from E.

## Antigenic role of NS2A beyond co-ancestry with E

To rule out sites that may have harbored antigenic signals from co-ancestry with E, we concatenated 60 random NS2A sites to the E protein sequence to adjust for its effects and reran the inference. Frequency at which a site has nonzero effect substitutions when included as part of the random subsets was summarized across 300 randomizations. The number of times sites were included in the subsets ranged from 60 to 107 with a median of 82. Considering 50% nonzero effect frequency as the null, an observed 80% frequency with these sample sizes would achieve a 1% false positive rate and <1% false negative rate under a binomial test. RMSE of E protein concatenated with NS2A sites was lower than when concatenated with sites outside of E and NS2A (Fig 5A). Changing the number of sites from 60 to 30 yielded consistent results (Fig 5B). The frequency at which sites in NS2A were estimated to have nonzero effects appeared bimodal, either showing effects in <1% of the number of times sampled or >99%, Fig 5C. When the 62 sites suggested by both analyses harboring signals beyond linkage with sites in E were concatenated to E, the effects remained (Fig 5D). Its performance (95%IQR of RMSE: 0.68, 0.70) was better than the performance using data concatenating E with random sets of 62 sites from other proteins (95%IQR: 0.71, 0.73), and the same sites but with residues permuted across viruses (lowest achieved 95%IQR: 0.72, 0.74). The within site permutation dissolves the association between residues and antigenicity of the virus but maintains the within site diversity. The consistently lower RMSE of the actual protein sequence supports the existence of antigenic signals in these NS2A sites beyond the linkage with E and was unlikely mapped to the antigenic patterns at random. The refitted substitution model with these 62 sites concatenated to the sequences of E was able to explain 54.2% to 54.5% of the variations in distances (S3(B) Fig), a 3.3% to 4.0% increase from E alone, with improvements more pronounced for interserotypic variations (5.6% to 6.6% increase) than intraserotypic (0.6% to 1.0% increase).

## Distributions of antigenic determinants in NS2A

The 62 sites identified in NS2A (S2 File) were scattered throughout the protein involving all eight predicted transmembrane segments of the protein [34], S6 Fig, covering 28.8% of NS2A sites: 31.2% of sites in the cytosol, 36.2% in the ER lumen, and 21.1% in the transmembrane domain. None of the protein segments included more nonzero effect sites than what would be expected if effects were assigned to variable sites at random (S7 Fig). The protein segment with

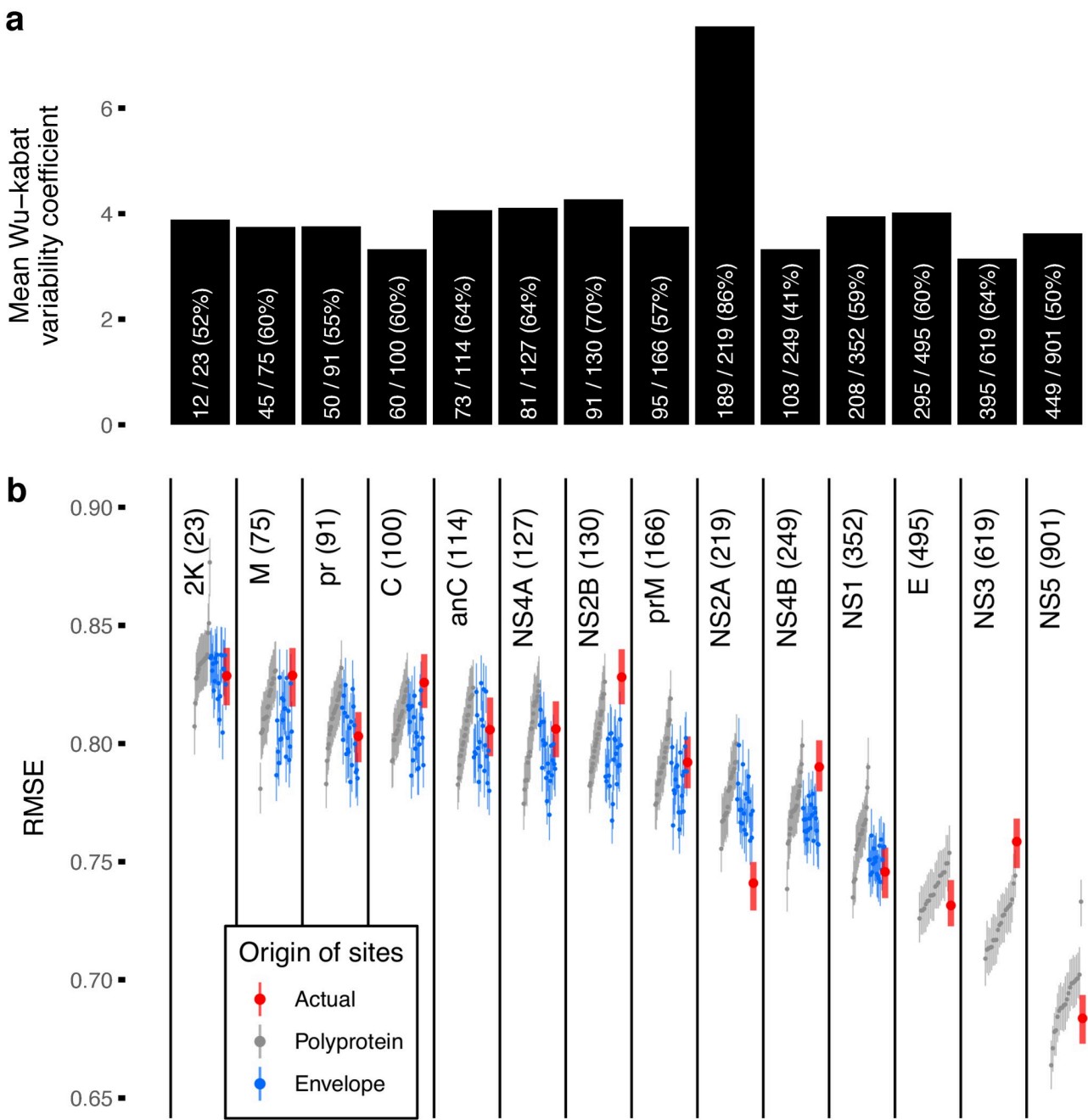

**Fig 4. Antigenic signal in each DENV protein.** a) Average within site variability in DENV proteins observed in the dataset. Bars were annotated with number of variable sites, total number of sites, and percentage of sites variable. b) Prediction performance of each DENV protein as observed (red) contrasted against expectations derived from random subsample of sites from any DENV protein of the same length (gray) and random down samples of sites from the envelope protein (E, blue). Points and lines are median and 95% interquartile range (IQR) of the root mean squared error (RMSE) evaluated under 100-fold Monte Carlo cross-validation. Length of the proteins are shown in parentheses. Only nonstructural protein 2A (NS2A) appeared to have better predictive performance than the expectations.

the lowest chance of observing this number of nonzero effect sites at random was pTMS-2 (p = 0.135, 9/20 sites, 45%). Segment pTMS-2 does not truly transverse the ER membrane but electrostatically interacts with the membrane through residues 30–55 [35]. With its characterized properties, Nemeśio and Villalaín [35] speculated that it has a role in membrane

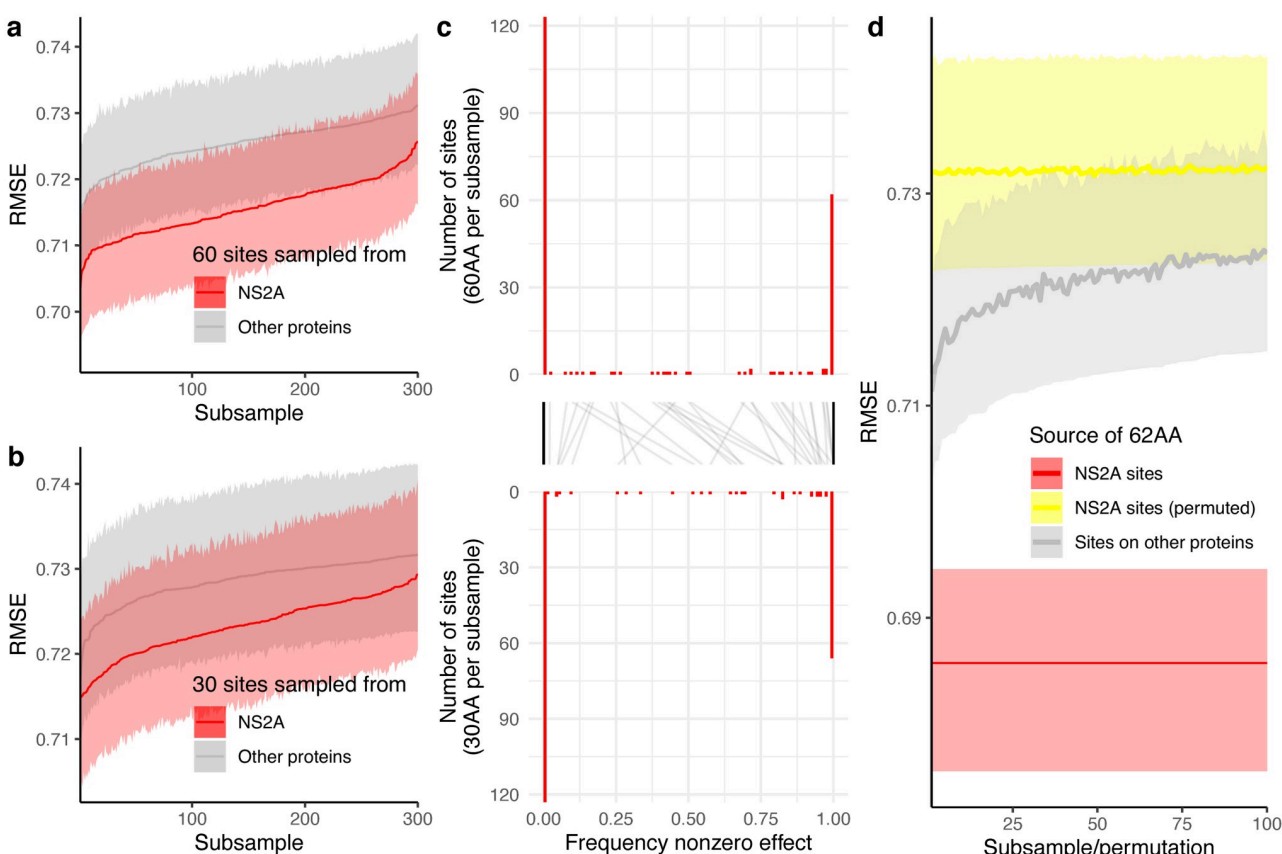

**Fig 5. Sites embedding antigenic signals beyond the envelope protein.** Prediction performance of downsampled NS2A sites concatenated with E when randomly downsampled to a) 60 sites and b) 30 sites contrasted against when concatenated with random sites from other proteins. c) Distribution of frequencies at which sites showed non-zero effect given being sampled in the two downsampling schemes. Black lines link frequencies of the same sites. d) Performance when concatenating the 62 sites which >99% of the times sampled was estimated to have non-zero effect size when adjusted for E in both schemes (red) compared against the same sites but permuted (yellow), and sites from other proteins of the same length (gray). Permutation was done by permuting residues observed at each site across viruses to conserve its diversity.

rearrangements during replication. Another region with a high proportion of sites harboring signals is the C-terminal which influences viral assembly and secretion [36]: 4/9 sites (44%), p = 0.337. It may be noteworthy that both sites preceding and after pTMS-5 where a predicted amphipathic helix resides [34] also showed nonzero effects. Like pTMS-2, pTMS-5 is located in the ER lumen and does not transverse the membrane, but was not found to be associated with the membrane [34]. The role of pTMS-5 is unclear but mutation D125A disrupted the signaling of NS1-NS2A cleavage abolishing the viral RNA synthesis [37]. Recent findings revealed how NS2A couples the encapsulation of viral RNA and the assembly of infectious virion [38]. NS2A binds prM in the C-prM-E polyprotein and the 3'UTR of the viral RNA. Following a cascade of cleavages, C transverses through the ER membrane, encapsulates the viral RNA, and the nucleocapsid buds with prM and E into the ER lumen. Our identified sites are not at the key molecules that modulate these activities but are proximally localized in the ER lumen. Like how a single amino acid change in a non-epitope site on the E protein could affect the conformation of the virus ensembled leading to differences in neutralization profiles [39], the influence of these sites on the interaction during the cascade of C-prM-E cleavage and assembly may have led to differences in resulting virions, and thus, vary their antigenicity.

## E-NS2A coevolution hotspots supports interprotein interactions

Coevolution between sites may indicate interprotein interactions [40]. To explore whether antigenic signals in NS2A could be linked to interactions with sites in E, we applied two coevolution detection methods to our dataset. The first method, *fastcov* [41], retains both site and residue information and takes into account asynchronous changes at different sites. The use of covariance between sites in the method has been shown to correspond well with branching patterns in the phylogeny. S8 Fig illustrates the density of coevolving residue pairs between sites in E and NS2A identified by *fastcov*. The second method, *SpydrPick* [42], is a mutual information (MI) based method with phylogenetic signal adjustment that detects coevolution between nucleotide positions. Filtering for pairs of nucleotide positions with MI values greater than the 99[th] percentile across all position pairs on the genome, NS2A appears to show a comparatively high level of coevolution with E compared to other proteins (S9 Fig). The coevolution hotspots suggested by both methods were around positions 40 (pTMS-2), 115 (pTMS-4), and 160 (pTMS-6) in the NS2A protein, which coincide with regions of high diversity and the identified nonzero effect sites. These results suggest possible interactions between E and NS2A at these sites.

## Effects of substitutions are background-specific

Drawing from the existing diversity of the 348 closely-related virus strains in our dataset, we examined whether the marginal effects identified in the substitution model could be observed for viruses separated by individual substitutions. We consider viruses with identical sequences in E and the 62 nonzero effect sites in NS2A as effectively identical. With the high genetic similarity between viruses in our dataset, we were able to identify pairs of viruses that were separated by a substitution of interest (virus $i$ vs. virus $j$) and a control virus that was otherwise effectively identical (virus $j^c$). We identified a sufficient number of these 'triplets' $(i, j, j^c)$ to test isolated effects of six substitutions in footprints of human-derived mAb (hmAb), one substitution in EDI/II/III but outside of known mAb epitopes, eight substitutions in stem/anchor domain of E, and twenty substitutions in NS2A. No nonzero effect substitutions in footprints of murine-derived mAb (mmAb) but outside of hmAb footprints had sufficient virus triplets for evaluation. The number of virus triplets were primarily limited by low number of control viruses due to coupling of the substitutions with other substitutions (470/698 substitutions). Notably, of the strongest effect sizes observed in our models, 138/229 substitutions with effect size >0.5 were not testable with the triplet analysis because these mutations were often accompanied by other antigenically important changes.

We found broad correspondence between differences in antigenic distances observed from virus triplets and effect sizes estimated by our model in all substitution groups (S10 Fig). For all substitutions, differences in antigenic distance observed from virus triplets ($\Delta D_m$) have wide 95%IQR. Given that we have matched for all changes in E and the 62 NS2A sites, we suspect that the wide confidence intervals are due to smaller sample sizes of 'testable' triplets. This validation is thus likely underpowered and cannot overcome variability of the measurements, an issue that would also likely affect experimental studies introducing individual mutations synthetically into infectious clones. We found that none of the 6 testable substitutions in footprints of hmAb had significant effects (S11 Fig). However, the genetic background had an important effect on the significance of each triplet. Take for example a substitution in the footprint of hmAbs on E, M160K, which has been shown experimentally to have a modest antigenic effect [43]). S12 Fig contrasts the overall distribution of $\Delta D_m$ for M160K against $\Delta D_m$ associated with each virus tested individually. Nearly half of the individual viruses have significant effects, and these effects are clustered when mapped to the phylogeny, indicating the effect

is dependent on the background genome (S13 Fig). This suggests that the particular virus this mutation is introduced into will affect the magnitude of the antigenic effect observed, even when working with closely related viruses of the same genotype circulating in a single city over time.

In the few substitutions that involved multiple serotype pairs, effects of the substitutions appeared to vary by serotype. For instance, albeit significant overall effects were observed for E:S169P, DENV2-DENV3 pairs were far from rejecting the null (S14 Fig). This heterogeneity further suggests that the effects of substitutions are background-dependent, which also partly explains the wide variation observed in $\Delta D_m$ pooled across virus triplets with variable backgrounds.

We also performed the triplet analyses on other sites in E and in NS2A. We found significant effects ($p \leq 0.05$) for 1 of 8 substitutions in EDI/II/III but outside of known mAb epitopes (S169P, S14 Fig), 0 of 1 substitutions in stem/anchor of E (S15 Fig) and 2 of 20 substitutions in NS2A (L19F and C41L, S16 Fig). The NS2A substitution C41L is in one of the coevolution hotspots with E, and is within pTMS-2, the region most associated with antigenic effect in our larger model.

## Discussion

Through studying viruses sampled from a concentrated locality over long periods of time, we were able to recover a large portion of known antigenically relevant sites targeted by neutralizing mAbs. The small number of substitutions separating the viruses within each serotype (and genotype) allowed the substitution model to more precisely draw the link between genetic and antigenic heterogeneity compared to past studies [23]. The fact that viruses of each serotype in the Thai dataset were mainly of a particular genotype means that these changes were associated with antigenic variation within genotypes, an aspect that has rarely been studied.

Our studies of the E protein suggest our model is likely conservative in attributing effects to sites/substitutions and is returning hits more specific to antibody responses in primates. Of the sites on the E protein marked as antigenically relevant by our model, 63.6% were within or neighborhooding known human epitopes but not mouse epitopes. This association was greater than random chance within 3 positions or 3.5 angstroms around known epitopes. Of the remaining antigenically relevant sites, 16/28 were in the stem/anchor domains, which have recently been shown to become exposed under physiological conditions but mAb targeting these sites have yet been identified. These comparisons provide support for antigenic signal in sites as measured by polyclonal responses, which may be similar to identified monoclonal antibodies but may target the same antigenic regions in a slightly different way. Alternatively, some of the sites we identified were not near known epitopes. Our findings suggest that polyclonal antisera may target epitopes beyond those of currently identified monoclonal antibodies and also support recent studies showing that changes at specific sites may introduce global changes to the virus that affect polyclonal neutralization in a non-epitope specific manner.

With the availability of whole genome sequences, we were able to assess the presence of antigenic signals in all DENV proteins. In doing so, we detected an excess of signal in NS2A. We further went on to identify the sites embedding the information in NS2A and observed a small gain in antigenic variation explained, especially distances between serotypes (5.6% to 6.6%), compared to only considering sites in E. Many of these sites were found to coincide with E-NS2A hotspots in our coevolution analyses. These findings suggest that changes in replication machinery (NS2A) in addition to changes in structural proteins (E) may influence antigenic properties of the virus. We did not find an association between other segments of the genome and antigenic change. Notably, we did not include untranslated regions (UTRs) in

our analyses, despite works that suggest their roles in replication [44]. However, the contributions, if any, may not be totally lost so long as linkage between sites in the coding sequence and the UTR are retained. Also, we did not account for recombinations between DENV, which has been reported to occur within serotypes between homologous sites [33]. Although this complicates phylogenetic reconstructions, our model is unlikely to be affected by recombination as it is phylogeny-free. In fact, presence of recombination accelerates the dissolve of linkage between sites, increasing diversity of sequence combinations, which makes effects of individual substitutions more likely to be detected.

An important part of our analysis approach is that we are able to query residues across all four serotypes and diverse circulating strains, enabling us to identify the effect of individual mutations within specific genetic backgrounds in an epidemiological context. To further evaluate how the observed effects hold across genetic backgrounds, we tested whether viruses with and without identified antigenic determinants in E and NS2A differ in antigenicity in the absence of other sources of antigenically relevant changes, thus drawing on the redundancy in our existing dataset to identify the antigenic effects attributable to single amino acid changes. These analyses are a prerequisite for experimental studies to test individual mutations in a clonal background. Our analysis shows that the background virus is important, suggesting experimental studies to identify substitutions driving antigenic changes should be conducted using infectious clones specific to the virus population under study. As designing infectious clones for flaviviruses is difficult, these substitutions provide the best candidates for extensive studies to uncover molecular mechanisms underlying the relationship between these substitutions and changes in antigenicity of DENV.

Measurement of the genetic determinants of antigenic difference will inform development of diagnostics to allow finer characterization of virus antigenic properties and establish the link between antigenic variation and severity of infection [45]. Further, identification of the genetic changes that contribute to antigenic variation is an enabling step towards the study of DENV evolution. Replacements of invading genotypes have primarily been attributed to substitutions that confer functional advantage, e.g., infectivity in specific cell types [46] and transmissibility [47, 48], which do not depend on past infection histories in the population. However, differences in susceptibility to heterotypic immunity have also been linked to clade replacements, suggesting antigenic selection may have a role in DENV evolution [49, 50]. In support of this hypothesis, we recently showed that antigenic traits of DENV have changed over time and are associated with both epidemic dynamics and genotype replacement [22]. The genetic determinants of antigenic differences identified in our study here will enable formal inferences into these evolutionary processes. Deeper exploration of DENV antigenic variation and factors underlying its evolution is needed to inform development of more broadly effective preventive and therapeutic countermeasures.

## Material and methods

### Ethics statement

This study was approved by the ethical review boards of the Queen Sirikit National Institute of Child Health, Walter Reed Army Institute of Research, and Johns Hopkins Bloomberg School of Public Health (former location of DATC) and University of Florida. The work of NIH and WRAIR was deemed non-human subjects research by their respective IRBs. Because researchers at UF, Cambridge and QSNICH can link these data to identifiers (age and location, though not used in this study), IRB approval was obtained at these institutions. These IRB approvals (Queen Sirikit National Institute of Children's Health (QSNICH 61–062), University of Florida (UF IRB201701844) and the University of Cambridge (HBREC.2019.36)) include a waiver

of consent. We followed the National Institutes of Health guidelines for the humane treatment of laboratory animals. The NIAID Animal Care and Use Committee approved these protocols (11DEN33 and 14DEN34, parent protocol NIAID ASP LID 9).

## Data

Our study utilized whole genome sequences and 3-dimensional antigenic map coordinates of 348 DENV previously characterized by Katzelnick et al [22]. In brief, 1,944 isolated viruses were derived from serum specimens collected from acute illnesses admitted to the Queen Siri-kit National Institute of Child Health (QSNICH) in Bangkok, Thailand, mostly between 1994 and 2014. Aside from a genotype replacement of DENV3 from genotype II to genotype III, viruses were primarily of a single dominant genotype for each serotype (DENV1 genotype I, DENV2 genotype Asian I, DENV4 genotype I). From the 1,944 whole genome sequences acquired (667 DENV1, 440 DENV2, 454 DENV3, and 383 DENV4), the isolates were system-atically sampled to represent amino acid variations in the envelope (E) protein and pre-mem-brane (prM) protein and to balance across all years between 1994 and 2014, resulting in 348 virus isolates (18%; 87 DENV1, 80 DENV2, 90 DENV3, and 91 DENV4) being antigenically characterized. Plaque reduction neutralization titers (PRNT) for a panel of twenty anti-DENV antisera were determined for all 348 viruses. The antisera were of *Chlorocebus sabaeus* chal-lenged with global representative DENV strains, five per serotype, as described elsewhere [3]. Using antigenic cartography, viruses were placed onto map coordinates in N-dimensions to best preserve the measured PRNT50 titers, finding 3-dimensions to be optimal. Euclidean dis-tances on the map are in units of $\log_2$ neutralization titer reductions. Antigenic data are stored on Zenodo (doi:10.5281/zenodo.5365818). To obtain pairwise antigenic distances between the viruses, we computed the Euclidean distances between their coordinates. Substitutions sepa-rating virus pairs were identified from translated coding sequences of each gene in the whole genome sequence alignment. All sequence data is publicly available on GenBank (accession numbers: KY586306 to KY586946, MW881266, MW945425 to MW945427, MW945430, MW945433 to MW945437, MW945454 to MW945763, MW945772 to MW946604, MW946607 to MW946985).

## Substitution effect size estimation

We adapted the substitution model described in Bell et al. [23] to analyze the data in our study.

$$D_{ij} \approx \hat{D}_{ij} = \sum_m d_m + v_i + v_j$$

Our model approximates the observed antigenic distance $D_{ij}$ between virus $i$ and virus $j$ to the predicted antigenic distance $\hat{D}_{ij}$. The predicted distance is a sum of effects of all substitu-tions present between the two viruses, $\sum_m d_m$ where $d_m$ is the effect of a single substitution $m$, and virus-specific measurement uncertainties, $v_i$ and $v_j$. For a pair of identical viruses, $\sum_m d_m = 0$, any antigenic distance observed between them equals to $v_i + v_j$. The use of map distances in our study intrinsically implies $D_{ij} = D_{ji}$. Nonetheless, we still follow the initial formulations and allowed the effects of the residue changes $m$ and its inverse to be asymmetric. L1 regulari-zation terms on $d_m$ were added to favor attributing the effects to a small number of substitu-tions. Virus-specific intercepts were under L2 regularization. Weights of these regularization terms were governed by three parameters which were set to the values used in Bell et al., $\lambda = 3.0$, $\kappa = 0.6$, $\delta = 1.2$, where the relatively high value of $\lambda$ disfavors attributing effects to substitu-tions, reducing the chance of falsely attributing effects to substitutions. Results were shown to be insensitive to these values [23]. Parameters were solved as quadratic programming

problems under nonnegativity constraints using the function *pnnqp* in R package *lsei* minimizing the following cost function.

$$C = \sum_{i,j} (\hat{D}_{ij} - D_{ij})^2 + \lambda \sum_m d_m + \kappa \sum_i v_i^2 + \delta \sum_j v_j^2$$

Effect size estimations were repeated 100 times, including random 90% of the virus pairs each time. The 10% held out were used to test the performance of each estimation. In each estimation, substitutions present in only one virus pair were excluded. To avoid collinearity, sites with the same residue patterns were grouped into clusters. Estimates were summarized by its median and 95% interquartile range (IQR). Substitutions were determined as having nonzero effects if the 95%IQR excluded zero. Root mean squared error (RMSE) evaluated using the test sets were used to describe the prediction performance of the fits.

### Performance of antigenic distance predictions

We evaluated the performance of the model separately for predicting antigenic distances based on mutations in the E protein, each DENV protein, each DENV protein concatenated to E, and within NS2A. For each of the 100 estimations, we predicted the antigenic distances for the 10% of virus pairs held out during the estimation process. To estimate predicted antigenic distances, where the virus specific intercepts are not known, we sum the effects of the substitutions separating them and adding twice the mean per-virus intercept to the sum. We compute the root mean squared error (RMSE) between predicted distances and antigenic distances derived from the 3-dimensional antigenic map. We report the median and 95% interquartile ranges (IQR) across the 100 estimations.

### Quantifying effects of measurement variability on antigenic distance

We synthesized 100 datasets with multiple sets of measurements of the sample virus included to study the amount of antigenic distance attributable to measurement variability of the PRNT50 assay. For each dataset, we randomly selected eight viruses (two per serotype) and synthesized four sets of PRNT measurements per each of these viruses by multiplying the original titers with scaling factors $10^m$, $m$ sampled from a normal distribution of mean one and variance 0.13 [51]. Together with observed measurements of the remaining viruses in the original dataset, 3-dimensional coordinates of the synthetic entries were inferred through antigenic cartography. We computed the pairwise distances between synthetic entries of the same viruses and divided the amount by two to quantify the contribution of each virus entry.

### Assessing sensitivity of effect determination threshold

Corrections for multiple comparisons involve adjusting the stringency of significance thresholds [52]. We counted the number of estimations that each substitution showed nonzero effect and divided the count by the number of estimations at which effect size estimation of the substitution was attempted to obtain the proportion of estimations in which substitutions showed nonzero effect. We examined the change in number of substitutions with significant effects as we increased the threshold proportion.

### Assessing association between effect sites and known epitopes

For a set of epitope neighborhood sites $M$ and a set of nonzero effect sites $S$, the observed number of overlap between them equals $|M \cap S|$. If $|S|$ nonzero effect sites were sampled from the set of variable sites $V$ at random, we would expect the proportion of overlap $p$ to be $\frac{|M \cap V|}{|V|}$.

Because effects can only be attributed to variable sites, $S \subset V$, it follows that $|S \cap V| = |S|$ and $|M \cap S \cap V| = |M \cap S|$. The binomial probability of observing an overlap of at least $|M \cap S|$ if $S$ was sampled from $V$ at random equals

$$\sum_{u=|M \cap S|}^{|S|} \binom{|S|}{u} p^u (1-p)^{|S|-u}$$

### Identifying E-NS2A coevolution hotspots

Coevolution analyses were performed using sequences of all 1,944 Thai viruses in our study. Alignment of E protein sequences concatenated with NS2A sequences was used as input for *fastcov v1.0.3* [41] with default configurations. Covarying residues passing default significance thresholds were extracted. *SpydrPick v1.2.0* [42] with *–linear-genome* flag was performed on whole genome alignments of the viruses. Pairs of nucleotide positions with mutual information (MI) values greater than $99^{th}$ percentile of MI values across all pairs were extracted. Density of extracted pairs by both methods were visualized using *geom_density_2d_filled* in R package *ggplot2* [53].

### Assessing observable isolated effects of substitutions

To evaluate further how the specific substitutions estimated to have nonzero effects by the substitution model hold across genetic backgrounds, a suitable first step is to test whether viruses with and without these substitutions differ in antigenicity in the absence of other sources of antigenically relevant changes. Thus, we queried our dataset for virus triplets to as closely simulate experimental validation using infectious clones, where each mutation would be introduced separately into clonal backgrounds. Because our outcome measure is antigenic distance, the equivalent experiment would be to take a reference virus $i$ and measure the fold-difference in titers across all sera in the serum panel to virus $j$. We would then do the same with control virus $j^c$, which is equivalent to virus $j$. All measures of distance are antigenic distances between pairs of viruses, which is related to the fold-drop in neutralization titers.

In our dataset of Thai DENV, we identified virus pairs $(i, j)$ that were separated a set of substitutions $M$ where the substitution of interest $m \in M$, then queried for control viruses $j^c$ where substitutions separating $(i, j^c)$ equals $M - \{m\}$. As a result of the common substitution requirement, $j$ and $j^c$ were always of the same serotype. For each virus triplet $(i, j, j^c)$ identified, we compute the difference in observed antigenic distance between $(i, j)$ and $(i, j^c)$. We denote this difference as $\Delta D_m$. In considering only substitutions in E and the 62 sites in NS2A, our analysis assumes that substitutions outside of E and the 62 NS2A sites do not contribute to antigenic changes. We derived the p-value in rejecting the null hypothesis that $\Delta D_m \leq 0$ by calculating the proportion of triplets with $\Delta D_m \leq 0$. As effects may be background dependent, the calculations were also done separately for each serotype pair of $(i, j)$ identified. Calculations were limited to sets of virus triplets that involved greater than two distinct viruses $i$ and had greater than 30 triplets identified.

### Supporting information

**S1 Fig. Time-calibrated maximum likelihood phylogenies of virus isolates.** Collected from Queen Sirikit National Institute of Child Health (QSNICH) between 1994–2014. Viruses selected for antigenic characterization were marked as orange circles.
(PDF)

**S2 Fig. Virus-specific intercepts fitted using E protein sequences.** a) Distributions and b) variation in virus-specific intercepts estimated using E protein sequences across the 100 estimations. Gray horizontal lines represent the mean intercepts across viruses for each of the estimations. c) Boxplot illustrating the amount of distance attributable to measurement variability across 100 synthetic samples. Divided by two to represent the per virus contribution. Thick lines denote the means (black) and medians (orange).
(PDF)

**S3 Fig. Relationship between observed antigenic distance and antigenic distance predicted by the substitution model.** a) when effects were fitted to envelope protein sequences (E) and b) when effects were fitted to E concatenated with 62 nonzero effect sites in nonstructural protein 2A (NS2A).
(PDF)

**S4 Fig. Association between effect sites and known epitopes of neutralizing antibodies.** a) Number and percentage of sites with and without effects by whether or not they are part of known epitopes. Odds ratios were calculated by either considering epitopes of both human-derived monoclonal antibodies (hmAb) and murine-derived monoclonal antibodies (mmAb) and when only restricted to hmAb epitopes. Defining neighborhoods of known epitopes as positions within N sites away (linear distance), the probability of nonzero effect sites being within the neighborhood at random (red) are contrasted against the proportion of variable sites that were within the neighborhood (gray): b) known epitopes for either hmAb or mmAb, c) known epitopes for hmAb, and d) known epitopes for mmAb outside of hmAb epitopes. N = 0 was when the neighborhood was exactly at the reported epitope positions. e, f, g) Respective analogous analysis but with neighborhoods defined as being within X angstroms away from known epitopes (3-dimensional spatial distance). X = 0 was when the neighborhood was exactly at the reported epitope positions.
(PDF)

**S5 Fig. Proportion of estimations in which substitutions showed nonzero effect.** a) Substitutions in envelope protein (E) only, ordered by the proportion at which substitutions showed nonzero effect across the 100 estimations. Substitutions identified by our threshold of 95% was highly similar to the maximum stringency of 100%; 372/394 substitutions (94.4%). Involvement was retained in 76/77 (99%) of the sites. b) In the analysis where E was concatenated to the 62 nonstructural protein 2A (NS2A) sites which consistently showed nonzero effects in our site sampling analysis, 292/304 substitutions (96.1%) in the NS2A sites remained nonzero at a threshold of 100%. Involvement was retained in 62/62 (100%) of the sites. Proportions corresponding to nonzero effect substitutions reported in our study (threshold of 95%) are colored red.
(PDF)

**S6 Fig. Substitutions with non-zero effect sizes in NS2A.** Median effect size of substitutions across the 100-fold Monte Carlo cross-validations shown as points, 95% interquartile range as whiskers. Points are colored by locations of the sites: ER lumen (green), transmembrane (yellow), or cytosol (blue). Locations of the sites and domain annotations were taken from [34].
(PDF)

**S7 Fig. Distribution of nonzero effect sites across NS2A segments.** a) Total number of sites in each segment (hollow), number of variable sites (filled black), and number of sites estimated to have nonzero effects (filled red). b) Probability that at least these number of nonzero effect sites were associated with the segments at random. Amino acid positions of the segments

shown in parentheses.
(PDF)

**S8 Fig. Density of coevolving residue pairs detected by *fastcov*.** Density values were scaled to maximum value of one. Distributions of nonzero effect substitutions (red) and site-specific Wu-Kabat variability coefficient (gray) of the respective proteins are shown on top (nonstructural protein 2A, NS2A) and side (envelope protein, E).
(PDF)

**S9 Fig. Density of coevolving nucleotide pairs detected by *SpydrPick*.** a) Density of nucleotide positions with mutual information (MI) values greater than $99^{th}$ percentile of MI values between pairs throughout the DENV genome. Density scaled to maximum value of one. Thin rectangle corresponds to coevolution relationship between E gene (y-axis) and sites throughout the genome. Thick rectangle highlights relationship between E gene and NS2A gene. b) Density plot expanding the highlighted region in panel (a).
(PDF)

**S10 Fig. Relationship between difference in antigenic distance observed in virus triplets and effect size estimates from the substitution model.** Shown separately for substitutions located in epitopes of human-derived monoclonal antibodies (hmAb), E domain I/II/III but outside of known epitopes (EDI/II/III), E stem/anchor domain, and nonstructural protein 2A (NS2A). Points are the medians of the observations/estimates. Lines are 95% interquartile ranges.
(PDF)

**S11 Fig. Effects of substitutions in footprints of human-derived mAb (hmAb).** Difference in antigenic distance observed between pairs of viruses separated by the specific substitution and antigenic distance observed in respective effectively identical viruses without the substitution (control viruses). Thick lines show median and 95% interquartile range (IQR) for triplets of all serotype pairs combined. Thin lines show the median and 95%IQR for each serotype pair identified.
(PDF)

**S12 Fig. Observable effects of substitution differ within the same serotype pair.** a) Distribution of difference in antigenic distance, $\Delta D_m$, for E:M160K substitution including all triplets with the same serotype pair (DENV2, DENV2) and the resultant p-value shown in comparison to b) median and 95% interquartile range of $\Delta D_m$ shown separately for each virus $j$ involved in the virus triplets and their respective p-values.
(PDF)

**S13 Fig. Distribution of virus-specific difference in antigenic distance on the phylogeny.** Median difference in antigenic distance, $\Delta D_m$, specific to each virus $j$ involved in the virus triplets shown in S12 Fig are colored on the phylogeny. Points are shown as solid circles for p-values $\leq 0.05$ and as hollow triangles otherwise.
(PDF)

**S14 Fig. Effects of substitutions in EDI/II/III but outside of known mAb epitopes.** a) Difference in antigenic distance observed between pairs of viruses separated by the specific substitution and antigenic distance observed in respective effectively identical viruses without the substitution (control viruses). Thick lines show median and 95% interquartile range (IQR) for triplets of all serotype pairs combined. Thin lines show the median and 95%IQR for each serotype pair identified. b) Distribution of difference in antigenic distance for substitution with p-

value $\leq$ 0.1 colored by serotypes of the virus pairs.
(PDF)

**S15 Fig. Effects of substitutions in the stem/anchor domain of E.** Difference in antigenic distance observed between pairs of viruses separated by the specific substitution and antigenic distance observed in respective effectively identical viruses without the substitution (control viruses). Thick lines show median and 95% interquartile range (IQR) for triplets of all serotype pairs combined. Thin lines show the median and 95%IQR for each serotype pair identified.
(PDF)

**S16 Fig. Effects of substitutions in nonstructural protein 2A (NS2A).** a) Difference in antigenic distance observed between pairs of viruses separated by the specific substitution and antigenic distance observed in respective effectively identical viruses without the substitution (control viruses). Thick lines show median and 95% interquartile range (IQR) for triplets of all serotype pairs combined. Thin lines show the median and 95%IQR for each serotype pair identified. b) Distribution of difference in antigenic distance for substitutions with p-value $\leq$ 0.1 colored by serotypes of the virus pairs.
(PDF)

**S1 File. Nonzero effect substitutions in envelope protein (E).**
(CSV)

**S2 File. Nonzero effect substitutions in nonstructural protein 2A (NS2A).**
(CSV)

## Acknowledgments

We thank the study participants and all individuals involved in the collection and isolation of virus samples which were used to generate data used in this study.

## Disclaimer

Material has been reviewed by the Walter Reed Army Institute of Research. There is no objection to its presentation and/or publication. The opinions or assertions contained herein are the private views of the author, and are not to be construed as official, or as reflecting the views of the U.S. Department of the Army, the U.S. Department of Defense, or the U.S. Government.

## Author Contributions

**Conceptualization:** Angkana T. Huang, Henrik Salje, Isabel Rodriguez-Barraquer, Derek J. Smith, Richard Jarman, Stephen S. Whitehead, Derek A. T. Cummings, Leah C. Katzelnick.

**Data curation:** Angkana T. Huang, Ana Coello Escoto, Nayeem Chowdhury, Christian Chávez, Wiriya Rutvisuttinunt, Irina Maljkovic Berry, Gregory D. Gromowski, Damon W. Ellison, Anthony R. Jones, Stefan Fernandez, Richard Jarman, Stephen S. Whitehead, Derek A. T. Cummings, Leah C. Katzelnick.

**Formal analysis:** Angkana T. Huang, Derek A. T. Cummings, Leah C. Katzelnick.

**Funding acquisition:** Henrik Salje, Derek J. Smith, Richard Jarman, Stephen S. Whitehead, Derek A. T. Cummings, Leah C. Katzelnick.

**Investigation:** Angkana T. Huang, Henrik Salje, Isabel Rodriguez-Barraquer, Derek J. Smith, Stephen S. Whitehead, Derek A. T. Cummings, Leah C. Katzelnick.

**Methodology:** Angkana T. Huang, Henrik Salje, Ana Coello Escoto, Nayeem Chowdhury, Christian Chávez, Wiriya Rutvisuttinunt, Irina Maljkovic Berry, Gregory D. Gromowski, Butsaya Thaisomboonsuk, Ananda Nisalak, Derek J. Smith, Stephen S. Whitehead, Derek A. T. Cummings, Leah C. Katzelnick.

**Project administration:** Henrik Salje, Ana Coello Escoto, Nayeem Chowdhury, Christian Chávez, Wiriya Rutvisuttinunt, Irina Maljkovic Berry, Gregory D. Gromowski, Chonticha Klungthong, Butsaya Thaisomboonsuk, Ananda Nisalak, Damon W. Ellison, Anthony R. Jones, Stefan Fernandez, Derek J. Smith, Richard Jarman, Stephen S. Whitehead, Derek A. T. Cummings, Leah C. Katzelnick.

**Resources:** Henrik Salje, Richard Jarman, Stephen S. Whitehead, Leah C. Katzelnick.

**Supervision:** Chonticha Klungthong, Butsaya Thaisomboonsuk, Damon W. Ellison, Anthony R. Jones, Stefan Fernandez, Derek J. Smith, Richard Jarman, Stephen S. Whitehead, Derek A. T. Cummings, Leah C. Katzelnick.

**Validation:** Angkana T. Huang, Henrik Salje, Bernardo Garcia-Carreras, Lin Wang, Chonticha Klungthong, Butsaya Thaisomboonsuk, Luke M. Trimmer-Smith, Derek A. T. Cummings, Leah C. Katzelnick.

**Visualization:** Angkana T. Huang, Derek A. T. Cummings, Leah C. Katzelnick.

**Writing – original draft:** Angkana T. Huang, Henrik Salje, Bernardo Garcia-Carreras, Irina Maljkovic Berry, Luke M. Trimmer-Smith, Stephen J. Thomas, Derek A. T. Cummings, Leah C. Katzelnick.

**Writing – review & editing:** Angkana T. Huang, Henrik Salje, Ana Coello Escoto, Bernardo Garcia-Carreras, Wiriya Rutvisuttinunt, Irina Maljkovic Berry, Gregory D. Gromowski, Lin Wang, Luke M. Trimmer-Smith, Damon W. Ellison, Stephen J. Thomas, Derek A. T. Cummings, Leah C. Katzelnick.

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
