## [Decision Letter · Decision Letter 0]

6 Jan 2022

Dear Dr. Katzelnick,

Thank you very much for submitting your manuscript "Beneath the surface: Amino acid variation underlying two decades of dengue virus antigenic dynamics in Bangkok, Thailand" for consideration at PLOS Pathogens. As with all papers reviewed by the journal, your manuscript was reviewed by members of the editorial board and by several independent reviewers. In light of the reviews (below this email), we would like to invite the resubmission of a significantly-revised version that takes into account the reviewers' comments.

The reviewers appreciated the attention to an important topic and highlighted the potential significance of the data. However, weaknesses were identified that need to be addressed prior to publication. As emphasized by reviewers 2 and 3, experimental validation of newly identified antigenic sites is needed to fully support the conclusions of the work. While important for both E and NS2A, such validation is critical to explain the unexpected role of NS2A in viral antigenicity suggested by the observational data.

We cannot make any decision about publication until we have seen the revised manuscript and your response to the reviewers' comments. Your revised manuscript is also likely to be sent to reviewers for further evaluation.

Sincerely,

Anice C. Lowen

Associate Editor

PLOS Pathogens

Ana Fernandez-Sesma

Section Editor

PLOS Pathogens

Kasturi Haldar

Editor-in-Chief

PLOS Pathogens

orcid.org/0000-0001-5065-158X

Michael Malim

Editor-in-Chief

PLOS Pathogens

orcid.org/0000-0002-7699-2064

The reviewers appreciated the attention to an important topic and highlighted the potential significance of the data. However, weaknesses were identified that need to be addressed prior to publication. As emphasized by reviewers 2 and 3, experimental validation of newly identified antigenic sites is needed to fully support the conclusions of the work. While important for both E and NS2A, such validation is critical to explain the unexpected role of NS2A in viral antigenicity suggested by the observational data.

Reviewer's Responses to Questions

**Part I - Summary**

Reviewer #1: Huang et al. use a dataset of paired full genome dengue virus sequences with PRNT assays to measure the antigenic distance between pairs of viruses using antigenic cartography, and modify a previously published model to estimate the effects of amino acid changes that contribute to those antigenic distances. They find that sites in the E protein that are within 3 amino acids away from previously identified antibody footprints may contribute to antigenicity, as well as sites in NS2A. The method the authors use to look at combinations of sites in NS2a while controlling for association with E is clever. This paper builds nicely off of previous reports of dengue antigenic dynamics, and provides new and interesting data about the roles of substitutions beyond the E protein in antigenic variation. Overall, I think that the results are sound and that the authors' findings are novel and interesting. However, the paper would benefit from more information in the Methods, and clarification of a few points throughout the manuscript to make the paper more readily understandable for the virology audience of this journal.

Reviewer #2: Huang et al. utilized a large dataset of genome sequences and antigenic information (Katzelnick et al. Science 2021), to evaluate the genetic determinants of dengue virus antigenic diversification. They found that 77 of 295 positions with residue variability in the E protein conferred antigenic effects, with only 22 of them (~28%) mapping to known epitopes, thus expanding the number of residues involved in antibody recognition/responses. This information is very interesting and could inform vaccine development. By examining the role of the other 9 dengue virus proteins, they found that the nonstructural (NS) protein NS2A presented a signal for the antigenic diversity detected at antibody level with neutralization assays. They performed different analyses and tested different hypotheses to show that the role of antigenic diversification of NS2A is not linked to similar ancestries on the genome. The groups collaborating on this study are leaders on dengue epidemiology and immunity and the data is of interest, but there is no real explanation on how this NS protein is involved on antibody recognition and virus neutralization. It has been shown that NS2A plays a role in virus RNA replication and potentially in the evasion of the interferon response, but there is no study showing that antibodies are directed to the NS2A from dengue virus, making it difficult to develop a coherent explanation for these results.

Reviewer #3: Huang et al., examine the genetic variation among dengue viruses (DENV) from historic samples in Thailand that span several years. The authors examine the relationships between amino acid residue variation within and outside the Envelope protein, a target of neutralizing antibodies, in potentially modulating the neutralizing activity of antibodies. Overall, the authors report several interesting, hypothesis-generating observations. However, their findings fall short as they do not validate any of their descriptive analyses.

**Part II – Major Issues: Key Experiments Required for Acceptance**

Reviewer #1: (No Response)

Reviewer #2: 1. The data involving NS2A on dengue virus antigenic diversification is interesting but speculative. The authors are from multiple established and well-funded laboratories and should be able to provide experimental data rather than “inviting assessment of these effects in vitro” or state that "it would be interesting to see how the effects compare when introducing these substitutions into other genetic backgrounds experimentally."

2. From those 55 residues that do not map on known epitopes but predicted by the model to be involved in antigenic diversity (Fig 2), authors found that “30 were within 3 sites from known epitopes” suggesting a potential role of antibody recognition. This reviewer could not find how the authors calculated the distance from residues mapping to known epitopes. Was this calculated using linear sequences or structural information from the E protein? Please provide a better description on the methods section on how this analysis was carried out. Linear sequence information might not be the best predictor of the role of these residues on antibody recognition as “distant” residues could be “close” when the structural information is considered. Again, authors have all the resources to provide experimental data to confirm whether those 55 residues that do not map on known epitopes are involved in antigenic diversity.

3. The association of the NS2A protein to antibody recognition and virus antigenic diversification could be linked to interactions between these two proteins, E and NS2A, at the replication level. Authors could use phylogenetic methods and additional sequences from other studies to determine whether this signal is associated to co-evolution.

Reviewer #3: 1. Have the authors validated any of the Envelope AA residues that are outside of the mAb footprints with respect to having an impact on neutralizing antibodies? Are there viral isolates available or are there recombinant viruses that can be used to validate some of their findings in neutralization assays with specific monoclonal antibodies? There are several hits from EDI and EDII that came up on their nonzero effect size. While the computational data is interesting and potentially compelling, it would be good to validate the data with these well characterized mAbs: 1F4, 14C10, 2D22, and 5J7, EDE1-2B2, and EDE1-2C8.

2. It’s unclear how amino acid residues in NS2 would modulate antigenicity of dengue viruses. While the authors show statistically significant data in their nonzero sum size model, and speculate in the discussion of likely mechanisms underlying these mutations and their interactions with capsid and prM, these amino acid residues need to be validated through the generation of mutant NS2 viruses to demonstrate if the reversion of the major NS2 hits have a differential phenotype in terms of 1) antibody immune evasion, 2) viral infectivity, or 3) global conformational changes in antigenicity.

3. Are the authors correcting for multiple comparisons in their statistical analyses? It’s not clear from their methods if this is being done. As some of their p values are borderline significant, I suspect they will not be significant after correcting for multiple comparisons as they should do for rigor.

**Part III – Minor Issues: Editorial and Data Presentation Modifications**

Reviewer #1: 1. I applaud the authors for the brevity of their manuscript. However, the methods section was quite short, and at times difficult to decipher exactly what the authors did. I would suggest adding the following pieces of information into the methods to help readers who are not familiar with Bell et al.

- The authors should define what the hyperparameters are and why they are set to those values.

- The authors should make clear why 10% of measurements are being withheld. I assumed that this was because 90% of the measurements were used as training data, leaving the remaining 10% as test data, but a sentence explicitly clarifying this would be helpful.

- I had to read the Methods section of Bell et al to fully understand their model, and I would guess that other readers of Plos Pathogens would need to do the same. In Bell et al, Dij is connected to dm, vi, and pj, which represent virus avidity, serum potentcy, and the titer drop between viruses. Seeing the explicit connection of Dij to these values made it easier to understand how the effects of each individual mutation was estimated in the model, and the authors should add it. Currently, it is difficult to figure out how each individual effect is being estimated, given that the only parameter present is Dij, which represents (as I understand it) the sum of all mutations' effects. I suggest the authors add more explicit definitions in their model, including the connection of Dij to dm, vi, and pj.

2. I was a tad confused in the manuscript about how exactly the predictions they perform were being done. From my understanding, the authors built these antigenic maps, then estimated the effects of individual amino acid changes on those distances. However, the authors then describe predicting antigenic distances. Does this mean that the authors estimated antigenic distances with antigenic cartography, then estimated the effects of each individual amino acid change using the modified Bell et al model, then used that information to predict the combined antigenic effect of all amino acids for the strains that did not have PRNT data? Did the authors do this separately for each individual protein sequentially? A paragraph in the methods about how exactly these predictions were done, on which strains, and using data from which genes/ORFs would be helpful.

3. For the last paragraph in the first section, there isn't any data shown. It would be good to add the actual data as a plot showing the correlation between models fitted to E and observed distances.

4. The authors write on line 76, "The model identified 394 nonzero effect substitutions positioned on 77 of the 295 sites...". Later, on line 85, they write "158 positions in the E protein contribute to epitopes of characterized anti-DENV mAbs while 336 positions...". Are the authors referring to amino acid sites in 1 part, and nucleotides in the other? Are they referring to different proteins? I was confused about why the denominator for the number of sites on E is different in these 2 sentences.

5. Figure 3 is quite blurry and a bit difficult to read. Figure 3d especially is difficult to interpret because all of the points are overlapping. Perhaps a histogram would help in showing the bimodal distribution? As is, every site looks the same, and it is impossible to distinguish how many sites have 0 vs. 1 effects.

6. In Figure 2c, how do the authors interpret that their model estimated 0 effects for 1/4 of the known epitopes? Similarly, it seems like their model was equally likely to estimate 0 vs. non-0 effects for known epitopes. Why do they think this is?

7. In sections 110-115, the authors describe that individual gene trees match full genome trees, which would make sense if there is little recombination in dengue viruses. It would be nice to explicitly acknowledge whether dengue viruses recombine, add a reference, and directly acknowledge how their test accounts for that.

Reviewer #2: 1. Huang et al. found 394 substitutions with nonzero effect on 77 residues from the E protein. Notably, only 22 of them (~28%) mapped to known epitopes. While this information is presented in Fig 2, the exact location is missing. This data is very interesting and could inform other studies, so authors should provide a supplementary table with the list of all those 77 residues and describe which ones map on known epitopes.

2. The model predicted that over 2/3 (52/74) of the residues that presented variability and mapped to known epitopes were not involved in antigenic differences (zero effect size, Fig 2). Please specify whether those residues are (mostly) associated to residues mapped with non-human mAbs.

3. Authors should provide a better description on how the 348 viruses isolated (18% from total) were selected for this study. This reviewer needed to go back to the recently published paper from this group (Katzelnick et al. Science 2021) to gather more information on the distribution of serotypes and genotypes for this study. This could be included as additional panel for Fig 1.

4. Figure S2 should be plotted to summarize the data. It is hard to determine what proteins of dengue are the most variable. Moreover, authors stated that “295 site on the E protein” presented residue diversity in the Thai dataset, however, it is difficult to determine that number from the current figure. Please considering including this (revised) data as part of the main manuscript.

5. Lines 96-97: I believe the authors meant “41 residues (78.8%)”

6. Lines 172-173: Please provide the reference.

Reviewer #3: 1. The first sentence in the first abstract is inaccurate: “Neutralizing antibodies are important correlates of protection against dengue virus (DENV) infections.” What is known is that neutralizing antibodies are associated with protection from severe DENV disease. However, it is not known if neutralizing antibodies can prevent subclinical viral infections that are asymptomatic. The authors should change sentence for factual accuracy or provide conclusive data that states otherwise.

PLOS authors have the option to publish the peer review history of their article (what does this mean?). If published, this will include your full peer review and any attached files.

Reviewer #1: No

Reviewer #2: No

Reviewer #3: No
---

## [Decision Letter · Decision Letter 1]

5 Apr 2022

Dear Dr. Katzelnick,

We are pleased to inform you that your manuscript 'Beneath the surface: Amino acid variation underlying two decades of dengue virus antigenic dynamics in Bangkok, Thailand' has been provisionally accepted for publication in PLOS Pathogens.

Best regards,

Anice C. Lowen

Associate Editor

PLOS Pathogens

Ana Fernandez-Sesma

Section Editor

PLOS Pathogens

Kasturi Haldar

Editor-in-Chief

PLOS Pathogens

orcid.org/0000-0001-5065-158X

Michael Malim

Editor-in-Chief

PLOS Pathogens

orcid.org/0000-0002-7699-2064

Reviewer Comments (if any, and for reference):

Reviewer's Responses to Questions

**Part I - Summary**

Reviewer #1: The authors have been quite responsive to reviewer comments. The additional analysis of restricting to human-derived mAbs is a nice one that helps explain the discordance between the model and the previously identified epitope sites. The triplet analysis is clever and a nice addition as well. All of my comments have been addressed, and I am happy to recommend the manuscript for publication. While reading, I found a couple of typos, which I will include here:

line 64: "Finally, probe our virus set..." I think this is missing a "we"?

line 77: "..distribution of estimated..." I think this should be either "distributions" or "the distribution"

Reviewer #2: Authors addressed most of my comments, but they failed to provide empirical evidence for two of the major comments: (i) how the NS2A protein affects virus antigenicity and (ii) whether the new sites described in the E protein play a role on resistance to neutralization. While it is true that the reverse genetic system for dengue viruses is intractable, this system has been successfully used to demonstrate how differences on the E protein affect overall neutralization titers (examples: Messer et al. J Virol. 2016:5090-5097; Messer et al. PLoS Negl Trop Dis. 2012:e1486). Nevertheless, while single point mutations is highly desirable to test the findings described here, the authors have the isolated viruses used for neutralization testing and they could have tested other biological properties in an attempt to explain their observations for the involvement of NS2A in antigenicity (e.g. differences in cellular binding [Lo et al. PLoS One. 2016:e0166474], replication kinetics, glycosylation, defective interfering particles abundance), and test viruses carrying different mutations in the E protein with a panel of mAbs to assess the role of the newly described E residues on dengue antigenicity. If any of these experiments are feasible, the authors should provide a better biological explanation in the discussion for their observations regarding the NS2A protein.

Reviewer #3: My comments and questions have been properly addressed and answered. I recommend that this manuscript be accepted and published without delay.

**Part II – Major Issues: Key Experiments Required for Acceptance**

Reviewer #1: (No Response)

Reviewer #2: (No Response)

Reviewer #3: none

**Part III – Minor Issues: Editorial and Data Presentation Modifications**

Reviewer #1: (No Response)

Reviewer #2: (No Response)

Reviewer #3: none

PLOS authors have the option to publish the peer review history of their article (what does this mean?). If published, this will include your full peer review and any attached files.

Reviewer #1: No

Reviewer #2: No

Reviewer #3: No

---

## [Editor Report · Acceptance letter]

27 Apr 2022

Dear Dr. Katzelnick,

We are delighted to inform you that your manuscript, "Beneath the surface: Amino acid variation underlying two decades of dengue virus antigenic dynamics in Bangkok, Thailand," has been formally accepted for publication in PLOS Pathogens.

Best regards,

Kasturi Haldar

Editor-in-Chief

PLOS Pathogens

orcid.org/0000-0001-5065-158X

Michael Malim

Editor-in-Chief

PLOS Pathogens

orcid.org/0000-0002-7699-2064